# Compact Representation of Uncertainty in Clustering

**Craig S. Greenberg** [1,2]  **Nicholas Monath**[1]  **Ari Kobren**[1]  **Patrick Flaherty**[3]

**Andrew McGregor**[1]  **Andrew McCallum**[1]

[1]College of Information and Computer Sciences, University of Massachusetts Amherst
[2]National Institute of Standards and Technology
[3]Department of Mathematics and Statistics, University of Massachusetts Amherst
`{csgreenberg,nmonath,akobren,mcgregor,mccallum}@cs.umass.edu`
`flaherty@math.umass.edu`

## Abstract

For many classic structured prediction problems, probability distributions over the
dependent variables can be efficiently computed using widely-known algorithms
and data structures (such as forward-backward, and its corresponding trellis for
exact probability distributions in Markov models). However, we know of no previ-
ous work studying efficient representations of exact distributions over clusterings.
This paper presents definitions and proofs for a dynamic-programming inference
procedure that computes the partition function, the marginal probability of a cluster,
and the MAP clustering—all exactly. Rather than the $N^{th}$ Bell number, these exact
solutions take time and space proportional to the substantially smaller powerset
of $N$. Indeed, we improve upon the time complexity of the algorithm introduced
by Kohonen and Corander [11] for this problem by a factor of $N$. While still
large, this previously unknown result is intellectually interesting in its own right,
makes feasible exact inference for important real-world small data applications
(such as medicine), and provides a natural stepping stone towards sparse-trellis
approximations that enable further scalability (which we also explore). In experi-
ments, we demonstrate the superiority of our approach over approximate methods
in analyzing real-world gene expression data used in cancer treatment.

## 1   Introduction

Probabilistic models provide a rich framework for expressing and analyzing uncertain data because
they provide a full joint probability distribution rather than an uncalibrated score or point estimate.
There are many well-established, simple probabilistic models, for example Hidden Markov Models
(HMMs) for modeling sequences. Inference in HMMs is performed using the forward-backward
algorithm, which relies on an auxiliary data structure called a trellis (a graph-based dynamic program-
ming table). This trellis structure serves as a compact representation of the distribution over state
sequences. Many model structures compactly represent distributions and allow for efficient exact or
approximate inference of joint and marginal distributions.

Clustering is a classic unsupervised learning task. Classic clustering algorithms and even modern
ones, however, only provide a point estimate of the "best" partitioning by some metric. In many
applications, there are other partitions of the data that are nearly as good as the best one. Therefore
representing uncertainty in clustering can allow one to chose the most interpretable clustering from
among a nearly equivalent set of options. We explore the benefits of representing uncertainty in
clustering in a real-world gene expression analysis application in the experiments section.

Representing discrete distributions can be rather challenging, since the size of the support of the distribution can grow extremely rapidly. In the case of HMMs, the number of sequences that need to be represented is exponential in the sequence length. Despite this, the forward-backward algorithm (i.e., belief-propagation in a non-loopy graph) performs exact inference in time linear in the size of the sequence multiplied by the square of the size of the state space. In the case of clustering, the problem is far more difficult. The number of clusterings of $N$ elements, known as the $N^{th}$ Bell number [2], grows super exponentially in the number of elements to be clustered. For example, there are more than a billion ways to cluster 15 elements. An exhaustive approach would require enumerating and scoring each clustering. We seek a more compact representation of distributions over clusterings.

In this paper, we present a dynamic programming inference procedure that exactly computes the partition function, the marginal probability of a cluster, and the MAP clustering. Crucially, our approach computes exact solutions in time and space proportional to the size of the powerset of $N$, which is substantially less than the $N^{th}$ Bell number complexity of the exhaustive approach. While the size of the powerset of $N$ is still large, this is a previously unknown result that on its own bears intellectual interest. It further acts as a stepping stone towards approximations enabling larger scalability and provides insight on small data sets as shown in the experiments section.

The approach works by creating a directed acyclic graph (DAG), where each vertex represents an element of the powerset and there are edges between pairs of vertices that represent maximal subsets/minimal supersets of one another. We refer to this DAG as a *cluster trellis*. The dynamic programs can operate in either a top-down or bottom up fashion on the cluster trellis, labeling vertices with local partition functions and maximum values. It is also possible to read likely splits and joins of clusters (see Appendix M), as well as marginals from this structure. These programs work in any circumstance where the energy of a cluster can be computed. We prove that our programs return exact values and provide an analysis of their time and space complexity.

This paper also describes an approach to approximating the partition function, marginal probabilities, and MAP inference for clustering in reduced time and space. It works by performing exact computations on a sparsified version of the cluster trellis, where only a subset of the possible vertices are represented. This is analogous to using beam search [17] in HMMs. We prove that our programs return exact values for DAG-consistent partitions and that the time and space complexity are now measured in the size of the sparse cluster trellis. When not in the text, proofs of all facts and theorems can be found in the Appendix.

We develop our method in further detail in the context of correlation clustering [1]. In correlation clustering, the goal is to construct a clustering that maximizes the sum of cluster energies (minus the sum of the across cluster energies), where a cluster energy can be computed from pairwise affinities among data points. We give a dynamic program that computes the energies of all possible clusters. Our approach proceeds in a bottom up fashion with respect to the cluster trellis, annotating cluster energies at each step. This all can be found in the Appendix.

Previous work has examined the related problem of computing MAP *k-clusterings* exactly, including dynamic programming approaches [8, 9, 22], as well as using fast convolutions [11]. Our method has a smaller runtime complexity than using these approaches for computing the MAP clustering and partition function for all possible partitions (irrespective of $k$). Further, none of this related work discusses how to reduce complexity using approximation (as we do in Section 4), and it is unclear how their work might be extended for approximation. The most closely related work [10] models distributions over clusterings using Perturb and MAP [16]. Unlike the Perturb and MAP approach, our work focuses on exact inference in closed form.

Being able to compactly represent probability distributions over clusterings is a fundamental problem in managing uncertainty. This paper presents a dynamic programming approach to exact inference in clustering, reducing the time complexity of the problem from super exponential to sub-quadratic in the size of the cluster trellis.

## 2 Uncertainty in Clustering

Clustering is the task of dividing a dataset into disjoint sets of elements. Formally,

**Definition 1.** *(Clustering) Given a dataset of elements, $\mathcal{D} = \{x_i\}_{i=1}^N$, a **clustering** is a set of subsets, $\mathcal{C} = \{C_1, C_2, \ldots, C_K\}$ such that $C_i \subseteq \mathcal{D}$, $\bigcup_{i=1}^K C_i = \mathcal{D}$, and $C_i \cap C_j = \emptyset$ for all $C_i, C_j \in \mathcal{C}$, $i \neq j$. Each element of $\mathcal{C}$ is known as a* cluster.

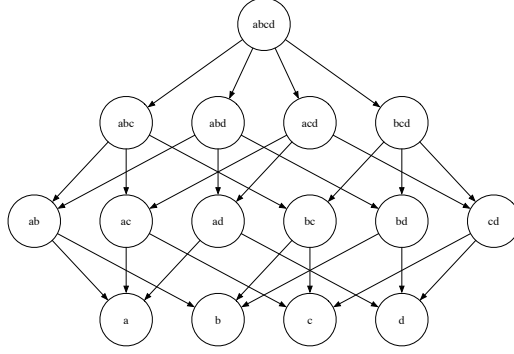

Figure 1: A cluster trellis, $\mathcal{T}$, over a dataset $\mathcal{D} = \{a, b, c, d\}$. Each node in the trellis represents a specific cluster, i.e., subset, of $\mathcal{D}$ corresponding to its label. Solid lines indicate parent-child relationships. Note that a parent may have multiple children and a child may have multiple parents.

Our goal is to design data structures and algorithms for efficiently computing the probability distribution over all clusterings of $\mathcal{D}$. We adopt an energy-based probability model for clustering, where the probability of a clustering is proportional to the product of the energies of the individual clusters making up the clustering. The primary assumption in energy based clustering is that clustering energy is decomposable as the product of cluster energies. While it is intuitive that the probability of elements being clustered together would be independent of the clustering of elements disjoint from the cluster, one could conceive of distributions that violate that assumption. An additional assumption is that exponentiating pairwise scores preserves item similarity. This is the Gibbs distribution, which has been found useful in practice [6].

**Definition 2.** *(Energy Based Clustering) Let $\mathcal{D}$ be a dataset, $\mathcal{C}$ be a clustering of $\mathcal{D}$ and $\mathcal{E}_{\mathcal{D}}(\mathcal{C})$ be the energy of $\mathcal{C}$. Then, the probability of $\mathcal{C}$ with respect to $\mathcal{D}$, $P_{\mathcal{D}}(\mathcal{C})$, is equal to the energy of $\mathcal{C}$ normalized by the partition function, $Z(\mathcal{D})$. This gives us $P_{\mathcal{D}}(\mathcal{C}) = \frac{\mathcal{E}_{\mathcal{D}}(\mathcal{C})}{Z_{\mathcal{D}}}$ and $Z(\mathcal{D}) = \sum_{\mathcal{C} \in \mathbb{C}_{\mathcal{D}}} \mathcal{E}_{\mathcal{D}}(\mathcal{C})$. The $\mathcal{E}_{\mathcal{D}}(\mathcal{C})$ energy of $\mathcal{C}$ is defined as the product of the energies of its clusters: $\mathcal{E}_{\mathcal{D}}(\mathcal{C}) = \prod_{C \in \mathcal{C}} \mathcal{E}_{\mathcal{D}}(C)$*

We use $\mathbb{C}_{\mathcal{D}}$ to refer to all clusterings of $\mathcal{D}$. In general, we assume that $\mathcal{D}$ is fixed and so we omit subscripts to simplify notation. Departing from convention [12], clusterings with higher energy are preferred to those with lower energy. Note that computing the membership probability of any element $x_i$ in any cluster $C_j$, as is done in mixture models, is ill-suited for our goal. In particular, this computation assumes a fixed clustering whereas our work focuses on computations performed with respect to the distribution over all possible clusterings.

## 3 The Cluster Trellis

Recall that our goal is compute a distribution over the valid clusterings of an instance of energy based clustering as efficiently as possible. Given a dataset $\mathcal{D}$, a naïve first step in computing such a distribution is to iterate through its unique clusters and, for each, compute its energy and add it to a running sum. If the number of elements is $|\mathcal{D}| = N$, the number of unique clusters is the $N^{th}$ Bell Number, which is super-exponential in $N$ [14].

Note that a cluster $C$ may appear in many clusterings of $\mathcal{D}$. For example, consider the dataset $\mathcal{D}' = \{a, b, c, d\}$. The cluster $\{a, b\}$ appears in 2 of the clusterings of $\mathcal{D}'$. More precisely, in a dataset comprised of $N$ elements, a cluster of $M$ elements appears in the $(N - M)^{th}$ Bell Number of its clusterings. This allows us to make use of memoization to compute the distribution over clusterings more efficiently, in a procedure akin to variable elimination in graphical models [4, 25]. Unlike variable elimination, our procedure is agnostic to the ordering of the elimination.

To support the exploitation of this memoization approach, we introduce an auxiliary data structure we call a *cluster trellis*.

**Definition 3.** *(Cluster Trellis) A cluster trellis, $\mathcal{T}$, over a dataset $\mathcal{D}$ is a graph, $(V(\mathcal{T}), E(\mathcal{T}))$, whose vertices represent all valid clusters of elements of D. The edges of the graph connect a pair vertices if one (the "child" node) is a maximal strict subset of the other (the "parent" node).*

In this paper, we refer to a cluster trellis simply as a *trellis*. In more detail, each trellis vertex, $v \in V(\mathcal{T})$, represents a unique cluster of elements; the vertices in $\mathcal{T}$ map one-to-one with the non-empty members of the powerset of the elements of $\mathcal{D}$. We define $\mathcal{D}(v)$ to be the elements in the cluster represented by $v$. There exists an edge from $v'$ to $v$, if $\mathcal{D}(v) \subset \mathcal{D}(v')$ and $\mathcal{D}(v') = \mathcal{D}(v) \cup \{x_i\}$ for some element $x_i \in \mathcal{D}$ (or vice versa). See Figure 1 for a visual representation of a trellis over 4 elements. Each vertex stores the energy of its associated cluster, $\mathcal{E}(\mathcal{D}(v))$, and can be queried in constant time. We borrow terminology from trees and say vertex $v'$ is a *parent* of vertex $v$, if there is an edge from $v'$ to $v$, and that vertex $v''$ is an *ancestor* of $v$ if there is a directed path from $v''$ to $v$.

### 3.1 Computing the Partition Function

Computing a distribution over an event space requires computing a partition function, or normalizing constant. We present an algorithm for computing the partition function, $Z(\mathcal{D})$, with respect to all possible clusterings of the elements of $\mathcal{D}$. Our algorithm uses the trellis and a particular memoization scheme to significantly reduce the computation required: from super-exponential to exponential.

The full partition function, $Z(\mathcal{D})$, can be expressed in terms of cluster energies and the partition functions of a specific set of *subtrellises*. A subtrellis rooted at $v$, denoted $\mathcal{T}[v]$ contains all nodes in $\mathcal{T}$ that are descendants of $v$.

Formally, a *subtrellis* $\mathcal{T}[v] = (V(\mathcal{T}[v]), E(\mathcal{T}[v]))$ has vertices and edges satisfying the following properties: (1) $V(\mathcal{T}[v]) = \{u | u \in V(\mathcal{T}) \wedge \mathcal{D}(u) \subseteq \mathcal{D}(v)\}$, and (2) $E(\mathcal{T}[v]) = \{(u, u') | (u, u') \in E(\mathcal{T}) \wedge u, u' \in V(\mathcal{T}[v])\}$. Note that $\mathcal{T}[v]$ is always a valid trellis.

The following procedure not only computes $Z(\mathcal{D})$, but also generalizes in a way that the partition function with respect to clusterings for any subset $\mathcal{D}(v) \subset \mathcal{D}$ can also be computed. We refer to the partition function for a dataset $\mathcal{D}(v)$ memoized at the trellis/subtrellis $\mathcal{T}[\mathcal{D}(v)]$ as the partition function for the trellis/subtrellis, $Z(\mathcal{T}[\mathcal{D}(v)])$.

---

**Algorithm 1** `PartitionFunction`$(\mathcal{T}, \mathcal{D})$

---

Pick $x_i \in \mathcal{D}$
$Z(\mathcal{D}) = 0$
**for** $v$ in $V(\mathcal{T})^{(i)}$ **do**
    Let $v'$ be such that $\mathcal{D}(v') = \mathcal{D} \setminus \mathcal{D}(v)$
    **if** $Z(\mathcal{D}(v'))$ has not been assigned **then**
        $Z(\mathcal{D}(v')) = $ `PartitionFunction`$(\mathcal{T}[v'], \mathcal{D}(v'))$
    $Z(\mathcal{D}) \leftarrow Z(\mathcal{D}) + \mathcal{E}(\mathcal{D}(v)) * Z(\mathcal{D}(v'))$
**return** $Z(\mathcal{D})$

---

Define $V(\mathcal{T})^{(i)} = \{v | v \in V(\mathcal{T}) \wedge x_i \in \mathcal{D}(v)\}$ and $\overline{V(\mathcal{T})^{(i)}} = V(\mathcal{T}) \setminus V(\mathcal{T})^{(i)}$. In other words, $V(\mathcal{T})^{(i)}$ is the set of all vertices in the trellis containing the element $x_i$ and $\overline{V(\mathcal{T})^{(i)}}$ is the set of all vertices that do not contain $x_i$.

**Fact 1.** *Let $v \in V(\mathcal{T})$ and $x_i \in \mathcal{D}(v)$. The partition function with respect to $\mathcal{D}(v)$ can be written recursively, with $Z(\mathcal{D}(v)) = \sum_{v_i \in V(\mathcal{T}[v])^{(i)}} \mathcal{E}(v_i) \cdot Z(\mathcal{D}(v) \setminus \mathcal{D}(v_i))$ and $Z(\emptyset) = 1$.*

*Proof.* The partition function $Z(\mathcal{D}(v))$ is defined as:

$$ Z(\mathcal{D}(v)) = \sum_{\mathcal{C} \in \mathbb{C}_{\mathcal{D}(v)}} \prod_{C \in \mathcal{C}} \mathcal{E}(C) $$

For a given element $x_i$ in $\mathcal{D}(v)$, the set of all clusterings of $\mathcal{D}(v)$ can be re-written to factor out the cluster containing $x_i$ in each clustering:

$$ \mathbb{C}_{\mathcal{D}(v)} = \{\{v_i\} \cup \mathcal{C} | v_i \in V^{(i)}, \mathcal{C} \in \mathbb{C}_{\mathcal{D}(v) \setminus \mathcal{D}(v_i)}\} $$

Note that $\mathbb{C}_{\mathcal{D}(v) \setminus \mathcal{D}(v_i)}$ refers to all clusterings on the elements $\mathcal{D}(v) \setminus \mathcal{D}(v_i)$. Using this expansion and since $\mathcal{E}(\{v_i\} \cup \mathcal{C}_i) = \mathcal{E}(\{v_i\})\mathcal{E}(\mathcal{C}_i)$, we can rewrite the partition function as below. By performing

algebraic re-arrangements and applying our definitions:

$$Z(\mathcal{D}(v)) = \sum_{v_i \in V^{(i)}} \sum_{\mathcal{C} \in \mathbb{C}_{\mathcal{D}(v) \setminus \mathcal{D}(v_i)}} \mathcal{E}(v_i)\mathcal{E}(\mathcal{C})$$

$$= \sum_{v_i \in V^{(i)}} \sum_{\mathcal{C} \in \mathbb{C}_{\mathcal{D}(v) \setminus \mathcal{D}(v_i)}} \mathcal{E}(v_i) \prod_{C \in \mathcal{C}} \mathcal{E}(C)$$

$$= \sum_{v_i \in V^{(i)}} \mathcal{E}(v_i) \sum_{\mathcal{C} \in \mathbb{C}_{\mathcal{D}(v) \setminus \mathcal{D}(v_i)}} \prod_{C \in \mathcal{C}} \mathcal{E}(C)$$

$$= \sum_{v_i \in V^{(i)}} \mathcal{E}(v_i) Z(\mathcal{D}(v) \setminus \mathcal{D}(v_i))$$

$\square$

As a result of Fact 1, we are able to construct a dynamic program for computing the partition function of a trellis as follows: (1) select an arbitrary element $x_i$ from the dataset; (2) construct $V(\mathcal{T})^{(i)}$ as defined above; (3) for each vertex $v_i \in V(\mathcal{T})^{(i)}$, compute and memoize the partition function of $\mathcal{D}(v) \setminus \mathcal{D}(v_i)$ if it is not already cached; (4) sum the partition function values obtained in step (3). The pseudocode for this dynamic program appears in Algorithm 1.

We use Algorithm 1 and Fact 1 to analyze the time and space complexity of computing the partition function. Consider a trellis $\mathcal{T}$ over a dataset $\mathcal{D} = \{x_i\}_{i=1}^N$. Our goal is to compute the partition function, $Z(\mathcal{T})$. When the partition function of all subtrellises of $\mathcal{T}$ have already been computed, Algorithm 1 is able to run without recursion.

**Fact 2.** *Let $\mathcal{T}$ be a trellis such that the partition function corresponding to each of its subtrellises $\mathcal{T}'$ is memoized and accessible in constant time. Then, $Z(\mathcal{T})$ can be computed by summing exactly $2^{N-1}$ terms. Given that the partition function of every strict sub-trellis of $\mathcal{T}$ (i.e., any sub-trellis of $\mathcal{T}$ that is not equivalent to $\mathcal{T}$) has been memoized and is accessible in constant time, then $Z(\mathcal{T})$ is computed by taking the sum of exactly $2^{N-1}$ terms.*

We now consider the more general case, where the partition function of all subtrellises of $\mathcal{T}$ have not yet been computed:

**Theorem 1.** *Let $\mathcal{T}$ be a trellis over $\mathcal{D} = \{x_i\}_{i=1}^N$. Then, $Z(\mathcal{T})$ can be computed in $\mathcal{O}(3^{N-1}) = \mathcal{O}(|V(\mathcal{T})|^{\log(3)})$ time.*

A proof of Theorem 1 can be found in the Appendix in Section E.

## 3.2 Finding the Maximal Energy Clustering

By making a minor alteration to Algorithm 1, we are also able to compute the value of and find the clustering with maximal energy. Specifically, at each vertex in the trellis, $v$, store the clustering of $\mathcal{D}(v)$ with maximal energy (and its associated energy). We begin by showing that there exists a recursive form of the max-partition calculation analogous to the computation of the partition function in Fact 1.

**Definition 4.** *(Maximal Clustering) Let $v \in V(\mathcal{T})$ and $x_i \in \mathcal{D}(v)$. The maximal clustering over the elements of $\mathcal{D}(v)$, $\mathcal{C}^\star(\mathcal{D}(v))$, is defined as: $\mathcal{C}^\star(\mathcal{D}(v)) = \operatorname{argmax}_{\mathcal{C} \in \mathbb{C}_{\mathcal{D}(v)}} \mathcal{E}(\mathcal{C})$.*

**Fact 3.** *$\mathcal{C}^\star(\mathcal{D}(v))$ can be written recursively as $\mathcal{C}^\star(\mathcal{D}(v)) = \operatorname{argmax}_{v' \in V(\mathcal{T}[v])^{(i)}} \mathcal{E}(v') \cdot \mathcal{E}(\mathcal{C}^\star(\mathcal{D}(v) \setminus \mathcal{D}(v')))$.*
In other words, the clustering with maximal energy over the set of elements, $\mathcal{D}(v)$ can be written as the energy of any cluster, $C$, in that clustering multiplied by a clustering with maximal energy over the elements $\mathcal{D}(v) \setminus C$.

Using this recursive definition, we modify Algorithm 1 to compute the maximum clustering instead of the partition function, resulting in Algorithm 2 (in Appendix). The correctness of this algorithm is demonstrated by Fact 3. We can now analyze the time complexity of the algorithm. We use similar memoized notation for the algorithm where $\mathcal{C}^\star(\mathcal{T}[\mathcal{D}(v)])$ is the memoized value for $\mathcal{C}^\star(\mathcal{D}(v))$ stored at $v$.

**Fact 4.** *Let $\mathcal{T}_\mathcal{D}$ be a trellis over $\mathcal{D} = \{x_i\}_{i=1}^N$. Then, $\mathcal{C}^\star(\mathcal{T}_\mathcal{D})$ can be computed in $\mathcal{O}(3^{N-1})$ time.*

## 3.3 Computing Marginals

The trellis facilitates the computation of two types of cluster marginals. First, the trellis can be used to compute the probability of a specific cluster, $\mathcal{D}(v)$, with respect to the distribution over all possible clusterings; and second, it can be used to compute the probability that any group of elements, $X$, are clustered together. We begin by analyzing the first style of marginal computation as it is used in computing the second.

Let $\mathbb{C}^{(v)} \in \mathbb{C}$ be the set of clusterings that contain the cluster $\mathcal{D}(v)$. Then the marginal probability of $\mathcal{D}(v)$ is given by $P(\mathcal{D}(v)) = \frac{\sum_{\mathcal{C} \in \mathbb{C}^{(v)}} \mathcal{E}(\mathcal{C})}{Z(\mathcal{D})}$, where $Z(\mathcal{D})$ is the partition function with respect to the full trellis described in section 2. This probability can be re-written in terms of the complement of $\mathcal{D}(v)$, i.e., $P(\mathcal{D}(v)) = \frac{\sum_{\mathcal{C} \in \mathbb{C}^{(v)}} \mathcal{E}(\mathcal{C})}{Z(\mathcal{D})} = \frac{\sum_{\mathcal{C} \in \mathbb{C}^{(v)}} \mathcal{E}(\mathcal{D}(v)) \mathcal{E}(\mathcal{C} \backslash \mathcal{D}(v))}{Z(\mathcal{D})} = \frac{\mathcal{E}(\mathcal{D}(v)) \sum_{\mathcal{C}' \in \mathbb{C}_{\mathcal{D} \backslash \mathcal{D}(v)}} \mathcal{E}(\mathcal{C}')}{Z(\mathcal{D})} = \frac{\mathcal{E}(\mathcal{D}(v)) Z(\mathcal{D} \backslash \mathcal{D}(v))}{Z(\mathcal{D})}$. Note that if $Z(\mathcal{D} \backslash \mathcal{D}(v))$ were memoized during Algorithm 1, then computing the marginal probability requires constant time and space equal to the size of the trellis. This is only true for clusters whose complements do not contain element $x_i$ (selected to compute $Z(\mathcal{D})$ in Algorithm 1), which is true for $|V(\mathcal{T})|/(2|V(\mathcal{T})| - 1)$ of the vertices in the trellis. Otherwise, we may need to repeat the calculation from Algorithm 1 to compute $Z(\mathcal{D} \backslash \mathcal{D}(v))$. We note that due to memoization, the complexity of computing the partition function of the remaining verticies is no greater than the complexity of Algorithm 1.

This machinery makes it possible to compute the second style of marginal. Given a set of elements, $X$, the marginal probability of the elements of $X$ being clustered together is: $P(X) = \sum_{\mathcal{D}(v) \in \mathcal{T}: X \subseteq \mathcal{D}(v)} P(\mathcal{D}(v))$. The probability that the elements of $X$ is distinct from the marginal probability of a cluster in that $P(X)$ sums the marginal probabilities of all clusters that include all elements of $X$. Once the marginal probability of each cluster is computed, the marginal probability of any sets of elements being clustered together can be computed in time and space linear in the size of the trellis.

## 4 The Sparse Trellis

The time to compute the partition function scales sub-quadratically with the size of the trellis (Theorem 1). Unfortunately, the size of the trellis scales exponentially with the size of the dataset, which limits the use of the trellis in practice. In this section, we show how to approximate the partition function and maximal partition of a *sparse trellis*, which is a trellis with some nodes omitted. Increasing the sparsity of a trellis enables the computation of approximate clustering distributions for larger datasets.

**Definition 5.** *(Sparse Trellis) Given a trellis* $\mathcal{T} = (V(\mathcal{T}), E(\mathcal{T}))$, *define a sparse trellis with respect to* $\mathcal{T}$ *to be any* $\widehat{\mathcal{T}} = (\widehat{V}, \hat{E})$ *satisfying the following properties:* $\widehat{V} \neq \emptyset$, $\widehat{V} \subset V(\mathcal{T})$, *and* $\hat{E} = \{(v, v') | \mathcal{D}(v') \subset \mathcal{D}(v) \wedge \nexists u : \mathcal{D}(v') \subset \mathcal{D}(u) \subset \mathcal{D}(v)\}$.

Note that there exist a number of sparse trellises that contain no valid clusterings. As an example, consider $\widehat{\mathcal{T}} = (\widehat{V} = \{v_1, v_2, v_3\}, \widehat{E} = \emptyset)$ where $\mathcal{D}(v_1) = \{a, b\}$, $\mathcal{D}(v_2) = \{b, c\}$, and $\mathcal{D}(v_3) = \{a, c\}$.

For ease of analysis, we focus on a specific family of sparse trellises which are closed under recursive complement [1]. This property ensures that the trellises contain only valid partitions. For trellises in this family we show that the partition function and the clustering with maximal energy can be computed using algorithms similar to those described in Section 3. Since these algorithms have complexity measured in the number of nodes in the trellis, their efficiency improves with trellis-sparsity. We also present the family of tree structured sparse trellises with tree specific partition function and max partition algorithms. The more general family of all sparse trellises is also discussed briefly.

The key challenge of analyzing a sparse trellis, $\widehat{\mathcal{T}}$, is how to treat any cluster $C$ that is not represented by a vertex $v \in \widehat{\mathcal{T}}$, i.e., $C = \mathcal{D}(v) \wedge v \notin \widehat{\mathcal{T}}$. Although there are several feasible approaches to reasoning about such clusters, in this paper we assume that any cluster that is not represented by a

vertex in $\widehat{\mathcal{T}}$ has zero energy. Since the energy of a clustering, $\mathcal{C}$, is the product of its clusters' energies (Definition 2), $\mathcal{E}(\mathcal{C}) = 0$ if it contains one or more clusters with zero energy.

## 4.1 Approximating The Partition Function and Max Partition

Given a sparse trellis, $\hat{\mathcal{T}}$, we are able to compute the partition function by using Algorithm 1.

**Fact 5.** *Let $\widehat{\mathcal{T}} = (\widehat{V}, \widehat{E})$ be a sparse trellis whose vertices are closed under recursive complement. Then Algorithm 1 computes $Z(\widehat{\mathcal{T}})$ in $\mathcal{O}(|\widehat{\mathcal{T}}|^{log(3)})$.*

If $\widehat{\mathcal{T}}$ is not closed under recursive complement, we cannot simply run Algorithm 1 because not all vertices for which the algorithm must compute energy (or the partition function) are guaranteed to exist. How to compute the partition function using such a trellis is an area of future study.

Given a sparse trellis, $\hat{\mathcal{T}}$, closed under recursive complement, we are able to compute the max partition by using Algorithm 2. Doing so takes $\mathcal{O}(|\hat{\mathcal{T}}|^{log(3)})$ time and $\mathcal{O}(|\hat{\mathcal{T}}|)$ space. The correctness and complexity analysis is the same as in Section 4.1.

The often-used hierarchical (tree structured) clustering encompasses one family of sparse trellises. Algorithms for tree structured trellises can be found in the Appendix in Section J.

## 5 Experiments

In this section, we demonstrate the utility of the cluster trellis via experiments on real-world gene expression data. To begin, we provide a high-level background on cancer subtypes to motivate the use of our method in the experiment in Section 5.3.

### 5.1 Background

For an oncologist, determining a prognosis and constructing a treatment plan for a patient is dependent on the subtype of that patient's cancer [13]. This is because different subtypes react well to some treatments, for example, to radiation and not chemotherapy, and for other subtypes the reverse is true [20]. For example, basal and erbB2+ subtypes of breast cancer are more sensitive to paclitaxel- and doxorubicin-containing preoperative chemotherapy (approx. 45% pathologic complete response) than the luminal and normal-like cancers (approx. 6% pathologic complete response)[18]. Unfortunately, identifying cancer subtypes is often non-trivial. One common method of learning about a patient's cancer subtype is to cluster their gene expression data along with other available expression data for which previous treatments and treatment outcomes are known [21].

### 5.2 Data & Methods

We use breast cancer transcriptome profiling (FPKM-UQ) data from The Cancer Genome Atlas (TCGA) because much is known about the gene expression patterns of this cancer type, yet there is heterogeneity in the clinical response of patients who are classified into the same subtype by standard approaches [23].

The data are subselected for African American women with Stage I breast cancer. We select African American women because there is a higher prevalence of the basal-like subtype among premenopausal African American women [15] and there is some evidence that there is heterogeneity (multiple clusters) even within this subtype [23]. Stage I breast cancer patients were selected because of the prognostic value in distinguishing aggressive subtypes from non-aggressive subtypes at an early stage.

Despite the considerable size of TCGA, there are only 11 samples meeting this basic, necessary inclusion/exclusion criteria. Each of the 11 samples is a 20,000 dimensional feature vector, where each dimension is a measure of how strongly a given gene is expressed. We begin by sub-selecting the 3000 features with greatest variance across the samples. We then add an infinitesimal value prior to taking the log of the remaining features, since genome expression data is believed to be normally distributed in log-space [19]. A similar data processing was shown to be effective in prior work [19].

We use correlation clustering as the energy model. Pairwise similarities are exponentiated negative euclidean distances. We subtract from each the mean pairwise similarity so that similarities are both positive and negative. We then compute the marginal probabilities for each pair (i.e., the probability

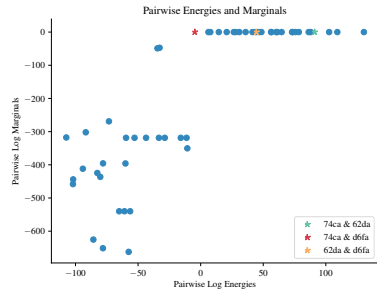

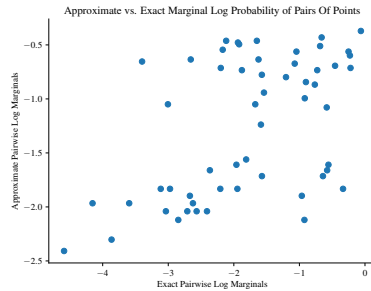

Figure 2: For each pair of patients with Stage I cancer, we plot the energy and marginal probability of the pair being in the same cluster as described in Section 5.3.

Figure 3: The approximate vs. exact pairwise marginals for each pair of gene expressions. Approximate marginals are computed using a Perturb-and-MAP based method [10].

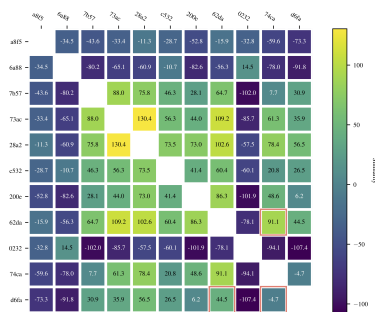

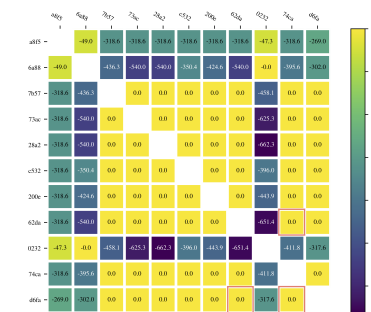

Figure 4: Heatmap of the pairwise energies between the patients. The pair `74ca` and `d6fa` has an energy of -4.7, `74ca` and `62da` have 91.09, and `d6fa` and `62da` have 44.5.

Figure 5: Heatmap of the marginal probability that a pair will be clustered together. Patients `74ca` and `d6fa` have a pairwise marginal that is nearly one, despite having a low pairwise energy.

that the two samples appear in the same cluster). See Section 3.3 for how to compute these values using the trellis.

## 5.3 Model Evaluation using Marginals

One method for evaluating a set of cancer subtype clustering models is to identify pairs of samples that the evaluator believes should be clustered together and inspect their pairwise energies. However, high pairwise energies do not necessarily mean the points will be clustered together by the model (which considers how the pairs' cluster assignment impacts the rest of the data). Similarly, a low pairwise energy does not necessarily mean the two samples will not be clustered together. The pairwise marginal on the other hand exactly captures the probability that the model will place the two samples in the same cluster. We test if the corresponding unnormalized pairwise energies or a simple approximation of the marginals could reasonably be used as a proxy for exact pairwise marginals.

### 5.3.1 Pairwise Energies vs. Marginals & Exact vs. Approximate Marginals

Figure 2 plots the pairwise log energies vs. pairwise log marginals of the sub-sampled TCGA data[2]. The pairwise scores and marginals are not strongly correlated, which suggests that unnormalized pairwise energies cannot reasonably be used as a proxy for pairwise marginals. For example, the sample pair of patients (partial id numbers given) `74ca` and `d6fa` have an energy of -4.7 (low), but a pairwise marginal that is nearly one. This is because both `74ca` and `d6fa` have high energy with sample `62da`, with pairwise energies 91.09 (the fourth largest) and 44.5, respectively. Figures 4 and 5 that visualize the pairwise energies and pairwise marginals, respectively.

We also explore the extent to which an approximate method can accurately capture pairwise marginals. We use an approach similar to Perturb-and-MAP [10]. We sample clusterings by adding Gumbel distributed noise to the pairwise energies and using Algorithm 2 to find the maximal clustering with the modified energies. We approximate the marginal probability of a given pair being clustered together by measuring how many of these sampled clusters contain the pair in the same cluster. Figure 3 plots the approximate vs. exact pairwise marginal for each pair of points in the dataset. The figure shows that the approximate method overestimates many of the pairwise marginals. Like the pairwise scores (rather than exact marginals), using the approximate marginals in practice may lead to errors in data analysis.

# 6  Related Work

While there is, to the best of our knowledge, no prior work on compact representations for exactly computing distributions over clusterings, there is a small amount related work on computing the MAP k-clustering exactly, as well as a wide array of related work in approximate methods, graphical models, probabilistic models for clustering, and clustering methods.

The first dynamic programming approach to computing the MAP k-clustering was given in [9], which focuses on minimizing the sum of square distances within clusters. It works by considering distributional form of the clusterings, i.e., all possible sizes of the clusters that comprise the clustering, and defines "arcs" between them. However, no full specification of the dynamic program is given and, as the author notes, many redundant computations are required, since there are many clusterings that share the same distributional form. In [8], the first implementation is given, with some of the redundancies removed, and the implementation and amount of redundancy is further improved upon in [22]. In each of these cases, the focus is on finding the best *k-clustering*, which can be done using these methods in $\mathcal{O}(3^n)$ time. These methods can also be used to find the MAP clustering for all $K$, however doing so would result in an $\mathcal{O}(n * 3^n)$ time, which is worse than our $\mathcal{O}(3^n)$ result.

In [11], the authors use fast convolutions to compute the MAP k-clustering and k-partition function. Fast convolutions use a Mobius transform and Mobius inversion on the subset lattice to compute the convolution in $\widetilde{\mathcal{O}}(n^2 2^n)$ time. It would seem promising to use this directly in our work, however, our algorithm divides the subset lattice in half,which prevents us from applying the fast transform directly. The authors note that, similar to the above dynamic programming approaches, their method can be used to compute the clustering partition function and MAP in $\mathcal{O}(n * 3^n)$, which is larger than our result of $\mathcal{O}(3^n)$. Their use of convolutions to compute posteriors of k-clusterings also implies the existence of an $\widetilde{\mathcal{O}}(n^3 2^n)$ algorithm to compute the pair-wise posterior matrix, i.e., the probability that items $i$ and $j$ are clustered together, though the authors mention that, due to numerical instability issues, using fast convolutions to computing the pair-wise posterior matrix is only faster in theory.

Recently proposed perturbation based methods [10] approximate distributions over clusterings as well as marginal distributions over clusters. They use the Perturb and MAP approach [16], originally proposed by Papandreou, which is based on adding Gumbel distributed noise to the clustering energy function. Unfortunately, for Perturb and MAP to approach the exact distribution, independent samples from the Gumbel distribution must be added to each clustering energy, which would require a super-exponential number of draws. To overcome this, Kappes et al. [10] propose adding Gumbel noise to the pairwise real-valued affinity scores, thus requiring fewer draws, but introducing some dependence among samples. They must also perform an outer relaxation in order obtain a computable bound for the log partition function. As a result, the method approaches a distribution with unknown approximation bounds.

# 7  Conclusion

In this paper, we present a data structure and dynamic-programming algorithm to compactly represent and compute probability distributions over clusterings. We believe this to be the first work on efficient representations of exact distributions over clusterings. We reduce the computation cost of the naïve exhaustive method from the $N^{th}$ Bell number to sub-quadratic in the substantially smaller powerset of $N$. We demonstrate how this result is a first step towards practical approximations enabling larger scalability and show a case study of the method applied to correlation clustering.

**Acknowledgments**

We thank the anonymous reviewers for their constructive feedback.

This work was supported in part by the Center for Intelligent Information Retrieval, in part by DARPA under agreement number FA8750-13-2-0020, in part by the National Science Foundation Graduate Research Fellowship under Grant No. NSF-1451512 and in part by the National Science Foundation Grant 1637536. Any opinions, findings and conclusions or recommendations expressed in this material are those of the authors and do not necessarily reflect those of the sponsor.

## Footnotes

[1]A set of sets, S, is closed under recursive complement iff $\forall x, y \in S, x \subset y \implies \exists z \in S : x \bigcup z = y \wedge x \cap z = \emptyset$.

[2]The MAP clustering of the dataset is in the Appendix in Section P.

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
