[Supplementary Material]

# Compact Representation of Uncertainty in Clustering
## Appendix

## A    Notation

| Symbol | Description |
|--------|-------------|
| $\mathcal{D}$ | a dataset of elements $x_1, \ldots, x_N$ |
| $x_i$ | an element of the dataset $\mathcal{D}$ |
| $C$ | a cluster of elements from $\mathcal{D}$ |
| $\mathcal{C}$ | a clustering, i.e. a set of clusters |
| $\mathbb{C}_{\mathcal{D}}$ | the set of all clusterings of $\mathcal{D}$ |
| $\mathcal{E}(C)$ | the energy of a cluster |
| $\mathcal{E}(\mathcal{C})$ | the energy of a clustering |
| $\mathcal{T}$ | cluster trellis with vertices $V$ and edges $E$ |
| $\mathcal{T}[v]$ | subtrellis rooted at $v$ |
| $\mathcal{D}(v)$ | cluster associated with node $v$ in a $\mathcal{T}$ |
| $V(\mathcal{T})^{(i)}$ | vertices in $\mathcal{T}$ containing $x_i$ |
| $\overline{V(\mathcal{T})^{(i)}}$ | vertices in $\mathcal{T}$ which do not contain $x_i$ |
| $Z(\mathcal{D}(v))$ | partition function w.r.t. $\mathcal{D}(v)$ |
| $Z(\mathcal{T}[v])$ | partition function memoized for $Z(\mathcal{D}(v))$ |

Table 1: Notation

## B    Case Study: Correlation Clustering

The energy-based clustering framework is compatible with any objective computed from a set of non-negative cluster scores[3].

One such objective that is widely used in practice is known as correlation clustering [1]. We present the traditional correlation clustering model in this section and present it in the energy based correlation clustering model in the next section. The input to correlation clustering is a complete (weighted) graph, $G = (V, E)$, where each edge has real-valued weight, i.e., $w_{uv} \in \mathbb{R}, \ (u,v) \in E$. The goal is to construct a clustering of the vertices that maximizes the sum of positive edge-weights within each cluster minus the negative edge-weights across the clusters.

Formally, let $\mathbb{C}_V$ be the set of all clusterings of $V$. Given a clustering $\mathcal{C} \in \mathbb{C}_V$, the sum of all positive within cluster edge-weights with respect to a clustering $\mathcal{C}$ is denoted $S^+(\mathcal{C}) = \sum_{C \in \mathcal{C}} \sum_{(u,v) \in C} w_{uv} \mathbb{1}_{\{w_{uv} > 0\}}$.

Similarly, $S^-(\mathcal{C})$ is the sum of the negative across-cluster edges with respect to $\mathcal{C}$. The optimal clustering $\mathcal{C}^\star \in \mathbb{C}_V$ is the one that maximizes the sum of positive within-cluster edge weights minus the sum of all negative across-cluster edge weights, $\mathcal{C}^\star = \max_{\mathcal{C} \in \mathbb{C}_V} S^+(\mathcal{C}) - S^-(\mathcal{C}) = \max_{\mathcal{C} \in \mathbb{C}_V} S^\pm(\mathcal{C})$. The problem is known to be NP-Hard[1].

There exist other objective functions over clusterings that are ordering-equivalent to $S^\pm(\cdot)$. Define $S(\mathcal{C}) = \sum_{C \in \mathcal{C}} \sum_{(u,v) \in C} w_{uv}$.

**Fact 6.** *Let $\mathcal{O}_{S^\pm}^\star(\mathbb{C}_{\mathcal{D}})$ be the sequence containing all clusterings of a set of elements, $\mathcal{D}$, in descending order with respect to $S^\pm(\cdot)$. Let $\mathcal{O}_S^\star(\mathbb{C}_{\mathcal{D}})$ be the sequence containing all clusterings of $\mathcal{D}$, in descending order with respect to $S(\cdot)$. Then $\mathcal{O}_{S^\pm}^\star(\mathbb{C}_{\mathcal{D}})$ and $\mathcal{O}_S^\star(\mathbb{C}_{\mathcal{D}})$ are ordering-equivalent.*

Fact 6 is not widely known, though it has occasionally been used implicitly. For example, Kappes et al [10], state $S(\cdot)$ as the correlation clustering objective. We provide a proof of Fact 6 in the Appendix.

Although the two methods for scoring a correlation clustering (i.e., $S^\pm(\cdot)$ vs. $S(\cdot)$) may compute different scores for the same clustering, Fact 6 implies that any clustering $\mathcal{C}$ of a dataset, $\mathcal{D}$, has the

same ordering under both objectives. Importantly, the best clustering, $\mathcal{C}^\star$, is equivalent under either objective. Our analysis focuses on $S(\cdot)$ since it is more convenient computationally when the number of clusters is large, which is common in practice [24].

The correlation clustering objective is computed in terms of positive and negative edge weights whereas our framework operates over non-negative energies. We can use a Gibbs distribution to transform cluster scores to energies, similar to [10]. Specifically, $\mathcal{E}(\mathcal{C}) = \prod_{C \in \mathcal{C}} \mathcal{E}(C) = \prod_{C \in \mathcal{C}} \exp[\sum_{(u,v) \in C} w_{uv}]$. After computing cluster energies, the full probability distribution over clusterings is constructed using the equations in Definition 2.

### B.1 Computing Cluster Energy

Computing the energy of cluster $C$ requires summing the $\frac{|C|(|C|-1)}{2}$ within-cluster edge weights. Since the number of potential clusters is $2^N - 1$ (for a dataset of size $N$), the naïve approach sums $\sum_{k=1}^{N} \binom{N}{k} \cdot \frac{|k|(|k|-1)}{2} = 2^{N-3}(N^2 - N) = O(N^2 \cdot 2^N)$ edge-weights.

**Fact 7.** *Let $C$ be a cluster with $|C| > 2$ and let $C_i$ and $C_j$ be two distinct clusters such that $C_i \subset C, C_j \subset C$ and $|C_i| = |C_j| = |C| - 1$. Then, the energy of $C$ can be expressed as $\mathcal{E}(C) = \frac{\mathcal{E}(C_i)\mathcal{E}(C_j)\mathcal{E}(C_i \backslash C_j \cup C_j \backslash C_i)}{\mathcal{E}(C_i \cap C_j)}$.*

Fact 7 follows from set algebra and the linearity of the energy function. Algorithm 3 exploits Fact 7 to speed up trellis construction. Algorithm 3 can be found in the Appendix.

**Fact 8.** *Algorithm 3 constructs a trellis, $\mathcal{T}$, for a graph $G = (V_G, E_G)$ and computes the energy of all clusters. Computing the energy of all clusters requires $\mathcal{O}(|V(\mathcal{T})|) = \binom{N}{2} + \sum_{k=3}^{N} \binom{N}{k} * 4$, steps where $N = |V_G|$.*

Specifically, cluster energies are memoized at each vertex in the trellis and then reused to compute the energies of new clusters before they are added to the trellis. The energy for cluster $C$ corresponding to vertex $v$ in the trellis is denoted $\mathcal{E}(v)$ in Algorithm 3. As described below, `ComputeEnergy` uses the fast computation described in Fact 7 with memoized values for each of the $\mathcal{E}$ terms at corresponding vertices in the trellis. A proof of Fact 8 is provided in the Appendix.

## C  Max Partition Algorithm

---
**Algorithm 2** `MaxCluster`$(\mathcal{T}, \mathcal{D})$

---
Pick $x_i \in \mathcal{D}$
MaxScore$(\mathcal{D}) \leftarrow 0$
MaxPart$(\mathcal{D}) \leftarrow \emptyset$
**for** $v$ in $V(\mathcal{T})^{(i)}$ **do**
    Let $v'$ be such that $\mathcal{D}(v') = \mathcal{D} \backslash \mathcal{D}(v)$
    **if** MaxScore$(\mathcal{D}(v'))$ has not been assigned **then**
        `MaxCluster`$(T[v'], \mathcal{D}(v'))$
        **if** MaxScore$(\mathcal{D}) < \mathcal{E}(\mathcal{D}(v)) \cdot$ MaxScore$(\mathcal{D}(v'))$ **then**
            MaxScore$(\mathcal{D}) = \mathcal{E}(\mathcal{D}(v)) \cdot$ MaxScore$(\mathcal{D}(v'))$
            MaxPart$(\mathcal{D}) = \mathcal{D}(v) \cup$ MaxPart$(\mathcal{D}(v'))$
**return** MaxPart$(\mathcal{D})$,MaxScore$(\mathcal{D})$

---

## D  Trellis Construction Algorithm

---

**Algorithm 3** ConstructTrellis($G = (V_G, E_G)$)

---

$V(\mathcal{T}) \leftarrow \emptyset$
$E(\mathcal{T}) \leftarrow \emptyset$
**for** $i \leftarrow 1$ through $|V_G|$ **do**
    **for** $C$ in $\mathbb{P}_i(V)$ **do**
        Let $v$ be a vertex corresponding to $C$
        $V(\mathcal{T}) \leftarrow V(\mathcal{T}) \cup \{v\}$
        **for** $C'$ in $\mathbb{P}_{i-1}(C)$ **do**
            Let $v'$ be the vertex corresponding to $C'$
            $E(\mathcal{T}) \leftarrow E(\mathcal{T}) \cup \{(v', v)\}$
        $\mathcal{E}(v) \leftarrow \texttt{ComputeEnergy}(v)$
**return** $\mathcal{T}$

---

## E  Proof of Theorem 1

*Proof.* We compute $Z(\mathcal{T})$ using the equation defined in Fact 1. To begin, an element $x_i \in \mathcal{D}$ is chosen and $V(\mathcal{T})^{(i)}$ is constructed. Computing $Z(\mathcal{T})$ requires the cluster energy of each $v \in V(\mathcal{T})^{(i)}$ (recall that there are $2^{N-1}$ such vertices) and the corresponding partition functions. These partition functions are computed for a sub-trellis $\mathcal{T}[v]$ such that $x_i \notin \mathcal{D}(v)$. Let $\mathcal{T}^k$ be the set of sub-trellises of $\mathcal{T}$ over $k$ elements none of which are $x_i$. If the partition function of every sub-trellis in $\mathcal{T}^{k-1}$ is computed and memoized before the partition function of any sub-trellis in $\mathcal{T}^k$, then, for any sub-trellis $\mathcal{T}[v] \in \mathcal{T}^k$, all relevant partition functions will have been memoized. By Fact 2, computing $Z(\mathcal{T}[v])$ includes exactly $2^{k-1}$ terms.

What remains to be analyzed is the number of sub-trellises in each set $\mathcal{T}^k$. Recall that any sub-trellis in $\mathcal{T}^k$ must not contain the element $x_i$. Then, the number of sub-trellises in $\mathcal{T}^k$ is $\binom{N-1}{k}$. Summing over all subtrellises, we must compute:

$$\sum_{k=1}^{N-1} \binom{N-1}{k} 2^{k-1} = \frac{1}{6}(3^N - 3)$$

terms. In total, computing the cost of the summing over all the subtrellis and the cost of computing $Z(\mathcal{T})$ as given by Fact 2 is $\frac{1}{6}(3^N - 3) + 2^{N-1} = \mathcal{O}(3^N)$. Since $|V(\mathcal{T})| = 2^N$, then $\mathcal{O}(3^N) = \mathcal{O}(|V(\mathcal{T})|^{\log(3)})$. $\qquad\square$

## F  Proof of Fact 2

*Proof.* According to Fact 1, $Z(\mathcal{T})$ can be written as a sum of products of cluster energies and partition functions. Note that $V(\mathcal{T})^{(i)}$ and $\overline{V(\mathcal{T})^{(i)}}$ are disjoint, $V(\mathcal{T})^{(i)} \cup \overline{V(\mathcal{T})^{(i)}} = V(\mathcal{T})$, and $\overline{V(\mathcal{T})^{(i)}}$ represents the nonempty sets in the powerset of $N-1$ elements. This implies the size of $V(\mathcal{T})^{(i)}$ is $2^{N-1}$. Therefore, in the special case described in Fact 2, the trellis can be used to compute the partition function in time $2^{N-1}$. $\qquad\square$

## G  Proof of Fact 3

*Proof.* Consider $\mathcal{C}^\star(\mathcal{D}(v))$, the clustering with the maximal energy over $\mathcal{D}(v)$. Select an arbitrary element $x_i \in \mathcal{D}(v)$. Since $\mathcal{C}^\star(\mathcal{D}(v))$ is a valid clustering, it must contain only one cluster that contains $x_i$; call that cluster $C_i^\star$. Let cluster $C_i^\star$ be represented by a node $v' \in V(\mathcal{T}[v])$. Given $C_i^\star$, we can construct $\mathcal{C}^\star(\mathcal{D}(v))$ by finding the maximal clustering of the remaining elements, i.e.,

$$\underset{v'' \in \mathbb{C}_{\mathcal{D}(v) \setminus \mathcal{D}(v')}}{\text{argmax}} \quad \mathcal{E}(\mathcal{D}(v'')) = \mathcal{E}(\mathcal{C}^\star(\mathcal{D}(v) \setminus \mathcal{D}(v')))$$

Finally, $C_i^\star \in \mathcal{C}^\star(\mathcal{D}(v))$ since we take the argmax with the respect to $V(\mathcal{T}[v])^{(i)}$ and $C_i^\star \in V(\mathcal{T}[v])^{(i)}$. $\qquad\square$

## H    Proof of Fact 4

*Proof.* Observe that the number of recursive calls to Algorithm 2 is the same as the number of recursive calls to Algorithm 1. Next, observe that the amount of computation required at each recursive call follows the same argument as in the proof of Theorem 1.     □

## I    Proof of Fact 5

*Proof.* Recall that Algorithm 1 begins by constructing $V(\mathcal{T})^{(i)}$ with respect to an arbitrary element $x_i$. Analogously, for the sparse trellis $\widehat{\mathcal{T}}$ we construct $\widehat{V}^{(i)}$. Note that $|\widehat{V}^{(i)}| < 2^{N-1}$ or else $\widehat{\mathcal{T}} = \mathcal{T}$. By our zero-energy assumption, for all $v \in V(\mathcal{T})^{(i)}\backslash\widehat{V}^{(i)}$, $\mathcal{E}(v) = 0$. Therefore, the energy of any clustering that is not computed by Algorithm 1 is also zero and may be omitted when computing the partition function of $\widehat{\mathcal{T}}$.

Finally, we show by contradiction that the algorithm does not omit any clusterings with non-zero energy. Assume that the algorithm does not compute the energy of a clustering $\mathcal{C}$ that has non-zero energy. Since $\mathcal{C}$ is a valid clustering, it must contain a cluster $C$ that contains the element $x_i$. If $C \in \widehat{\mathcal{T}}$ then the vertex $v$ that represents $C$ must be in $\widehat{V}^{(i)}$ and $\mathcal{E}(v) > 0$, so the algorithm would have computed its energy. Therefore, $v \notin \widehat{V}^{(i)}$ which means that $\mathcal{E}(C) = 0$, a contradiction.

By Theorem 1, this algorithm runs in $\mathcal{O}(|\widehat{\mathcal{T}}|^{log(3)})$ time and space linear in $|\widehat{\mathcal{T}}|$. Note that due to the constraint that $\widehat{\mathcal{T}}$ be closed under recursive complement, for every vertex $v$ in the sparse trellis such that $v$ has no parents in $\widehat{\mathcal{T}}$, each element $x_i \in \mathcal{D}(v)$ is contained in $|\widehat{\mathcal{T}}[v]|/2$ clusters, allowing the same counting argument as in the proof of Theorem 1.     □

## J    Tree-structured Sparse Trellis

The often-used hierarchical (tree structured) clustering encompasses one family of sparse trellises. Each vertex in a tree-structured trellis has at most one parent. A tree-structured trellis meets the definition of a sparse trellis since for any two vertices, $v_1, v_2$ in such a trellis, exactly one of the following must hold: $\mathcal{D}(v_1) \subset \mathcal{D}(v_2), \mathcal{D}(v_2) \subset \mathcal{D}(v_1)$, or $\mathcal{D}(v_1) \cap \mathcal{D}(v_2) = \emptyset$. This family has the advantage that many practical algorithms can be used for trellis construction, such as hierarchical agglomerative clustering. Previous work explores using trees to encode distributions over clusterings, though the focus is limited to modeling mixtures of tree consistent partitions rather than computing the marginals, maximal clusterings, and the partition function[7, 3].

We are able to compute the maximal clustering in a tree-structured (sparse) trellis, $\mathcal{T}$, in $\mathcal{O}(|\hat{\mathcal{T}}|)$ time and space as follows. Starting at the leaves, compare the energy of a parent and the product of its childrens' energies. Store the maximum of these two options at the parent, along with the corresponding clustering (either the parent or the union of the clusterings stored at each child). Continue the process upwards until the root of the trellis is reached. At the end of this process the root contains the clustering with the maximal energy as well as the corresponding energy. The proof of correctness is analogous to the one given for Fact 1.

We can use a similar technique to compute the partition function of a tree-structured trellis.

**Fact 9.** *Let $p$ be a parent vertex and let `ch(p)` be $p$'s children. Beginning at the leaves, proceed up the tree computing $Z(p) = \mathcal{E}(p) + \prod_{c \in ch(p)} Z(c)$, where $Z(\cdot)$ is the (memoized) partition function at a node. Then $Z(root)$ will contain the partition function for tree-consistent partitions.*

*Proof.* We must compute the partition at $p$. Note that if $p$ is a leaf, the partition function at $p$ is $\mathcal{E}(p)$. Otherwise, let `ch(p)` be the children of $p$. First, note that a valid clustering of $\mathcal{D}(p)$ consist of the union of a clustering from each of the children of $p$, $c \in ch(p)$. The same is true for $c$, and all of $p$'s descendents. Also, note that the energy of this sampled clustering is simply the product of the energies of the samples. Recall that a partition function of a child $v \in ch(p)$, $Z(v)$ is a sum of clustering energies over all clusterings of the descendants of $v$. Therefore, $\prod_{v' \in ch(p)} Z(v')$ is a sum of terms, each term being a product of one clustering from each child in `ch(p)`, i.e., a valid clustering of the descendants of $p$. Notice that this product contains *all* valid clusterings of the descendants of $p$

such that each valid clustering is built by taking the union of a clustering from each child in `ch(p)`. Also, no clustering will be double counted because all of $p$'s children are disjoint. Adding the energy of the complete partition, $\mathcal{E}(p)$, to this product computes the partition function of $p$. $\square$

# K  Proof of Fact 3

*Proof.* The algorithm constructs the trellis by computing clusters in ascending order of size– that is all clusters of size $i$ are computed before clusters of size $i + 1$. Edges are constructed between vertices $u$ and $v$ if $u$ represents a maximal proper subset of $v$'s cluster. Note that we use $\mathbb{P}_i(V)$ to represent the sets in the powerset of $V$ with cardinalty $i$.

Since the trellis is connected it is clear that every vertex will be visited. The sets represented by leaf nodes in the trellis are singletons, so there are no pairwise weights, therefore each leaf has score equal to 0 and energy equal to 1. The score of a vertex with two elements is defined by the correlation cluster affinity matrix. For all other verticies, $v$, we can select two children of $v$, call them $v_i, v_j$, such that $\mathcal{D}(v_i) \cup \mathcal{D}(v_j) = \mathcal{D}(v)$. We then compute $\mathcal{E}(\mathcal{D}(v))$ using Fact 7. Note that because $|\mathcal{D}(v_i)|$, $|\mathcal{D}(v_j)|$, $|\mathcal{D}(v_i) \cap \mathcal{D}(v_j)|$, and $|(\mathcal{D}(v_i) \setminus \mathcal{D}(v_j)) \cap (\mathcal{D}(v_j) \setminus \mathcal{D}(v_i))|$ are all less than $|\mathcal{D}(v)|$, each of their energies are computed by Algorithm 3 prior to computing $\mathcal{E}(\mathcal{D}(v))$ and need not be recomputed. Therefore the computation takes $\binom{N}{2} + \sum_{k=3}^{N} \binom{N}{k} * 4 = 4(2^N - N^2/2 - N/2 - 1) + 1/2(N-1)N = \mathcal{O}(2^N) = \mathcal{O}(|V(\mathcal{T})|)$ operations. $\square$

# L  Proof of Fact 6

Two functions, $f_1, f_2$ are said to be order equivalent iff $\forall c_i, c_j \in C, f_1(c_i) <= f_1(c_j) \implies f_2(c_i) <= f_2(c_j)$.

$$f(c) := \sum_{c_i \in C} \sum_{u,v \in C_i} w_{uv} * \mathbb{1}_{\{w_{uv} > 0\}}$$
$$- \sum_{c_i, c_j \in C} \sum_{u \in c_i} \sum_{v \in c_j} w_{uv} * \mathbb{1}_{\{w_{uv} <= 0\}}$$
$$g(c) := \sum_{c_i \in C} \sum_{u,v \in C_i} w_{uv}$$

We want to show that $f(c)$ is order equivalent with $g(c)$

*Proof.* Note that

$$\forall C \in \mathbb{C}_n, \quad \sum_{c_i, c_j \in C} \sum_{u \in c_i} \sum_{v \in c_j} w_{uv} * \mathbb{1}_{\{w_{uv} <= 0\}}$$
$$+ \sum_{c_i \in C} \sum_{u,v \in C_i} w_{uv} * \mathbb{1}_{\{w_{uv} <= 0\}} = E^-$$

where $E^-$ is a constant function of the input affinity matrix.

$$g(c) = \sum_{c_i \in C} \sum_{u,v \in C_i} w_{uv} * \mathbb{1}_{\{w_{uv}>0\}}$$

$$+ \sum_{c_i \in C} \sum_{u,v \in C_i} w_{uv} * \mathbb{1}_{\{w_{uv}<=0\}}$$

$$g(c) - \sum_{c_i \in C} \sum_{u,v \in C_i} w_{uv} * \mathbb{1}_{\{w_{uv}<=0\}} = f(c)$$

$$+ \sum_{c_i,c_j \in C} \sum_{u \in c_i} \sum_{v \in c_j} w_{uv} * \mathbb{1}_{\{w_{uv}<=0\}}$$

$$g(c) = f(c) + \sum_{c_i,c_j \in C} \sum_{u \in c_i} \sum_{v \in c_j} w_{uv} * \mathbb{1}_{\{w_{uv}<=0\}}$$

$$+ \sum_{c_i \in C} \sum_{u,v \in C_i} w_{uv} * \mathbb{1}_{\{w_{uv}<=0\}}$$

$$g(c) = f(c) + E^-$$

$\square$

## M    Likely Joins and Splits

The trellis can facilitate computing the most likely *join*, i.e., the pair of clusters that maximally increases the resultant clustering energy when combined:

$$\underset{C,C' \in \mathcal{C}}{\operatorname{argmax}}\{\mathcal{E}(C \cup C') * \mathbb{1}_{\{\mathcal{E}(C \cup C')>\mathcal{E}(C)*\mathcal{E}(C')\}}\}.$$

We also consider the most likely *split* of a cluster. For a cluster $C$, this reduces to finding the MAP estimate for the partitioning of $C$. This can be computed directly using the subtrellis rooted at the vertex corresponding to $C$. An alternative approach to using the trellis for finding likely splits for clusters in a clustering is to choose to place a restriction on the set of possible splits, for example that only a single datapoint can be split from an existing cluster $C \in \mathcal{C}$. These splits are the set of children of the verticies in the trellis corresponding to clusters in $\mathcal{C}$, and the one among them that maximally increases the resultant clustering energy is

$$\underset{u \in children(v(C))}{\operatorname{argmax}} \{\mathcal{E}(\mathcal{D}(u)) * \mathbb{1}_{\{\mathcal{E}(\mathcal{D}(u))>\mathcal{E}(\mathcal{D}(v))\}} | C \in \mathcal{C}\}$$

where

$$children(v) = \{u | u \in \mathcal{T}, \mathcal{D}(u) \subset \mathcal{D}(v),$$
$$||\mathcal{D}(u)| - |\mathcal{D}(v)|| = 1\}$$

and $v(C)$ is the vertex in $\mathcal{T}$ corresponding to $C$.

## N    Synthetic Data Example

We provide the following synthetic data example. We provide the probabilities and energies of various clusterings of a `Grid` dataset, in which energies are computed by correlation clustering and exponentiating the negative Euclidean distance between examples (which are simply evenly spaced points on a grid). Notice that the MAP clustering and other clusterings in the Grid dataset exhibit relatively similar probabilities.

(a) `Grid` dataset.

| | | | | |
|---|---|---|---|---|
| **MAP** | **Horizontal** | **Complete** | **Shattered** | **Inside** |

(b) Clustering probabilities.

| Data | Clustering | Prob. | Energy |
|------|-----------|-------|--------|
| | MAP | 2.262e-06 | 10.206 |
| | Horizontal | 5.403e-07 | 2.437 |
| Grid | Complete | 2.216e-07 | 1.000 |
| | Shattered | 2.216e-07 | 1.000 |
| | Inside | 7.968e-08 | 0.356 |

Figure 6: Probability of clusterings of the `Grid` dataset.

# O  Pairwise Energies vs. Marginals in UCI Zoo Dataset

We repeat our experiment comparing pairwise energies vs. marginals, as described in section 5, using data selected from the UCI zoo dataset [5]. Energies are computed by exponentiated cosine similarity. Figures 7 and 8 show that the energies and marginals for many pairs are not well correlated. For example, the pair sea wasp and termite has high energy, however the marginal probability of the pair being clustered together is low.

Figure 7: Heatmap of the pairwise energies between the animals.

Figure 8: Heatmap of the marginal probability that a pair of animals will be clustered together.

# P  MAP Clustering of TCGA

The MAP clustering of the TCGA subsample is
**C1**: {7b57, 74ca, 28a2, 73ac, 200e, 62da, d6fa, c532}, **C2**: {6a88, 0232}, **C3**: {a8f5}

## Footnotes

[3]One approach to using the energy-based clustering framework with negative cluster scores is to exponentiate the cluster scores prior to inputing them into the framework, as we will see in this section.