[Reviews · NeurIPS 2018]

Reviewer 1



This paper proposed to compute the distribution over all possible clusterings of a given set of samples. First, a cluster trellis was defined, where 1) each vertex represented one element from the power set of all samples and 2) vertex A was the parent of vertex B if A can be constructed as the union of A and one sample. Second, the partition function w.t.t. all possible clusterings can be computed recursively, resulting in exponential time complexity. Similarly, the clustering associated with the maximal energy can also be computed in a recursively manner. The time complexity was still exponential. In order to reduce the exponential complexity of the naive implementation of cluster trellis, the sparse trellis was introduced by removing most vertices from the trellis. One example of sparse trellis is that introduced by hierarchical clustering. Experimental results on a TCGA dataset consisting of 11 samples and 3000 features were presented. The exact pairwise marginals were compared against the approximate marginals. The idea of computing the exact distribution over all possible clusterings of the sample set seemed to be a novel idea. The introduction of cluster trellis and the recursive algorithms to compute the partition function and the optimal cluster were also novel. However, I'm not sure about the practicality of the proposed algorithm. First, the full cluster trellis has exponential complexity. For the sparse trellis, the complexity is determined by the number of vertices in the trellis. However, it's not clear how to determine the sparsity of the sparse trellis. Second, since this work emphasized the importance of computing the exact distribution over all possible clusterings, the introduction of the sparse trellis would inevitably compromise this objective because the remaining clusterings had to be determined by some heuristic methods, such as hierarchical clustering. Third, the experimental evaluation seemed limited because only one dataset was used and there were only 11 samples. More extensive evaluations on the synthetic and benchmark datasets, such as those datasets from the UCI repository, are needed.

Reviewer 2



Post-rebuttal: thank you for clarifying / correcting the figure with experimental results. Main idea: Exactly compute the partition function of a multicut / correlation clustering problem using dynamic programming. Strengths: Signed graph partitioning is an extremely versatile tool, so any theoretical or practical progress is welcome and may (eventually) have large impact. Studying a Gibbs distribution over solutions rather than just the minimal energy / MAP solution is of fundamental interest with a view to passing meaningful uncertainty estimates on to downstream processing. In essence, I think this is meaningful work. The paper is clearly written and thorough (though I did not check all mathematical details). Overall, I think the work has more substance than many NIPS papers I see, which is why I count it among the better 50%. Weaknesses: The method is of limited practical usefulness, with exact computations limited to 25 elements or so. The comparison with approximate methods (notably Kwikcluster) is not really fair. The state of the art to compare to is probably a set of multicut MAP solutions obtained on noisy versions of the original edge weights (ref. 7). This is not difficult to implement in practice and should give better results. Comments: I don't think the title summarizes the contribution well. There is not really much study or discussion of uncertainty, beyond the use of marginals. While marginals are a compact representation, the trellis certainly is not. Similarly, section 2 is entitled "Uncertainty in clustering" even though that topic is not discussed there at all. The core result which makes dynamic programming applicable is fact 1, proved in Appendix E. This being at the very heart of the contribution, I would recommend moving an extended version of Appendix E into the main text, at the expense of other parts. For instance, the MAP solution (section 3.2) which does not recur to branch & bound, is unlikely to be competitive with SOA ILP solvers in conjunction with efficient column generation. In fact, I wouldn't mind having the parts on sparse trellises omitted altogether. Without detailed analysis of what choice of trellis gives what kind of approximation quality, this section is not so useful. Bottom line: Fact 1 is all-important for this paper, and merits more extensive explanation / discussion in the main text. To give a better feeling for the data used, it would be good to show both the MAP clustering as well as all the marginals (in an 11x11 matrix). Fig 3: One would expect 11*(11-1)/2 = 55 entries, but there are fewer points in the scatter plot? Minor comments: Changing the definition of energy such that higher is preferable to lower (line 97) is not a good idea. The data used in the experiments is not a very convincing example for correlation clustering (coordinates in some space, with repulsive interactions obtained "artifically"; in particular, this choice induces a spherical shape bias on the resulting clusters). typo: elemnts line 241: unresolved reference Note on my confidence score: I did not check the maths; and I cannot guarantee that dynamic programming has not been used previously to compute the exact partition function of clusterings. I am confident of my other comments.

Reviewer 3



[Post-rebuttal: The authors' response addressed some of my concerns and I have no problem seeing this paper accepted. However, although the authors pointed out some differences between their method and the nonparametric Bayesian methods, they didn't address directly whether the nonparametric Bayesian methods could find an approximate probability of the clustering solutions. If those methods could, it would be better if the proposed method could be compared to those methods.] The paper proposed a method based on dynamic programming for computing the exact probability of clustering. The proposed method reduces the time complexity by a factor of N (number of data samples) compared to the best known complexity. The paper is generally well written. It studies a meaningful problem attempting to find the exact distribution of clustering. Though the resulting time complexity is still exponential in N and the proposed method is only feasible on small data sets, the reduction by a factor of N from the previous best known result looks significant. One limitation of the theoretical result is that it is based on the assumption of an energy-based probability model for clustering. Although the computation is exact, the resulting clustering distribution is still a approximation if the real data does not match the assumption of the energy model very much. I think the paper should make the implication of this assumption clear or discuss more about it. Even better, the authors may study the significance of this assumption using some synthetic data with known cluster labels in a longer version of this paper. The empirical study showed that a direct measure (pairwise similarity) is not as good as the proposed method for finding the probability that two data points belong to the same cluster (pairwise marginals). However, it does not seem to show that it is useful for finding such “pairwise marginals” or finding the distribution of clustering in practice. Questions: * There are some non-parametric clustering methods (e.g. Dirichlet Processes) that can give a (approximate) distribution over clustering models. Could you please explain the difference between your proposed method and those Bayesian methods in your paper? Minor comments: Line 81-82: “reducing the time complexity of the problem from super exponential to sub-quadratic in the size of the cluster trellis” => This comparison is misleading, since it is using a different units. Line 112: The paper relates the proposed method with Variable Elimination. Although I am quite familiar with VE, I don’t see many similarity between the proposed method and VE, except that both are based on dynamic programming. Perhaps the authors may elaborate on this. Algorithm 1 should move to top or bottom. It doesn’t have a proper location now. The authors may briefly explain why Fact 1 is true (in addition to the proof already provided) since this seems to be an important fact for the proposed method. Line 292: “This highlights the superiority” => The authors may explain how to see this.